# Winding Design and Efficiency Analysis of a Nine-Phase Induction Machine from a Three-Phase Induction Machine

**Ariel Fleitas** [1,*], **Magno Ayala** [2], **Osvaldo González** [2], **Larizza Delorme** [2], **Carlos Romero** [2], **Jorge Rodas** [2] and **Raul Gregor** [2]

1 Electrical Engineering Department, Universidad del Cono Sur de las Américas, Asuncion 1535, Paraguay
2 Laboratory of Power and Control Systems (LSPyC), Facultad de Ingeniería, Universidad Nacional de Asunción, Luque 2060, Paraguay
* Correspondence: ariel.fleitas162@gmail.com; Tel.: +595-985532964

**Abstract:** Multiphase machines are a hot research topic in control theory and industrial applications such as electric cars. However, the availability to buy them in the market is limited or null. For this reason, it is common to rewind it from a three-phase commercial machine. In this context, the aim of this paper is two-fold. First, to introduce a straightforward procedure to rewind a nine-phase induction machine from a three-phase one. For that purpose, a study of the three-phase induction motor was performed, which included selecting a new winding design, calculating stator coils, and simulating tests with ANSYS Maxwell software to validate the design. Secondly, a performance analysis comparing the power losses through experimental tests performed to obtain the electrical parameters of both nine-phase and three-phase topologies is presented.

**Keywords:** asymmetrical configuration; induction motors; motor rewinding; multi-phases machines





## 1. Introduction

The green agenda has directly influenced the research for higher efficiency of all-electric machines design and applications. Even though three-phase machine configuration has been and is still the standard choice for industrial applications, in the past decades, machines with a higher number of phases ($q > 3$), namely multiphase machines, have appeared as an exciting alternative [1]. Multiphase machines have several advantages compared with classical ones, such as better current/power per phase distribution [2], lower torque ripple than three-phase machines [3] and intrinsic fault-tolerance capabilities [4]. This is why multiphase systems can be found in electric boats, electric vehicles, railway traction, more electric aircraft, and wind power generation systems [3,5]. Some commercial examples include Dana six-phase and nine-phase electric trucks [6], and, in Ref. [7], a three-phase dual induction machine was selected as a suitable candidate for a belt-driven starter generator designed for 48 V mini-hybrid propulsion systems, and most recently, electrified propulsion engineering consultancy, Drive System Design (DSD) has developed a power-dense Electric Drive Unit (EDU) designed for a global Tier 1 supplier. Based on six-phase permanent magnet machines, the power-train will be used in a range of medium to heavy-duty commercial vehicle applications, such as city buses and delivery vans, when it reaches the market in 2024 [8]. Another application is "The USS Zumwalt", a US Navy full-electric propulsion ship powered by General Electric [9].

However, since multiphase machines are relatively new in the market, they lack mass production. Consequently, they must be developed for specific applications, usually from the stator windings of a commercial three-phase machine [10]. It should be noted that multiphase drive systems used for these machines protect against failures since, by increasing the number of phases ($q > 12$), the possibility of a failure occurring is 12% compared to a conventional three-phase system [11]. Still, with the increase in the number of phases, there is a greater degree of freedom for the operation against faults, with open

winding being the most common problem. The absorbed current is distributed to the other phases when losing a phase. Regarding copper losses, as the number of phases increases, a decrease in copper losses was demonstrated [11].

Concerning the design process of multiphase machines, a method for designing stator windings with any number of phases was proposed in [12]. A procedure for evaluating the parameters of a five-phase induction motor, which can also be applied to other multiphase motors with a higher number of phases, is presented in [13]. Meanwhile, ref. [14] describes an approach to design a six-phase motor, and in [15], specific problems related to the design of motors with a different number of phases are summarized, and results of the analysis are verified experimentally on a nine-phase asynchronous motor test bench. In ref. [16], a six-phase starting generator is presented and compared with a three-phase model. Similarly, ref. [17] compares, using finite element analysis, the performance of a nine-phase induction motor (IM) against a three-phase motor.

Even though several papers reported the winding design for multiphase machines, they are explained in a general way. This work proposes a straightforward procedure for constructing nine-phase IM stator windings from a three-phase IM without special requirements. The primary purpose is to help researchers who want to get involved in the multiphase machines field to quickly design a nine-phase topology, which falls within the category of multi-three-phase machines. A comparison between the efficiency of the commercial three-phase motor and the developed nine-phase redesign is made to obtain the reduction of power losses due to the windings quantitatively.

The rest of this work is organized as follows, Section 2 describes the characteristics of the three-phase motor, and Section 3 shows the analysis and design of the nine-phase IM. Then, Section 4 presents the experimental results, and Section 5 summarizes the conclusions.

## 2. Multi-Three-Phase Machines

Multi-three-phase machines have several independent sets to produce a unified electromagnetic field. The term *"sets"* is adopted to designate each three-phase winding it has. Therefore, a multi-three-phase machine consists of two or more three-phase winding and construction parameters are determined considering the number of stator slots ($K$), the number of poles ($2p$), and the number of phases ($q$) that these machines will have.

### 2.1. Multi-Three-Phase Winding

Multi-three-phase machines are generally made up of three-phase sets of three phases separated by 120° from each other. The stator winding arrangement is defined in the same way as for three-phase IM, that is, by the winding to be used, whether it concentric or eccentric, or if has one or two layers [14].

Regardless of the type of machine (synchronous or induction), multiphase machines can be classified into two well-differentiated groups: machines with a prime number of phases and machines with an even or odd number of phases.

- If the machine has a prime number of phases (3, 5, 7, 11, . . .), every two consecutive phases will be offset by $\frac{2\pi}{q}$ and will have a single isolated neutral point;
- If the machines have an even or odd number of phases (6, 9, 12, . . .), they can have $h$ sets of windings, each with $q$ phases and can have one or $h$ isolated neutral points. If the windings are three-phase sets ($q = 3$), the machine under analysis is a multi-three-phase IM ($qh$).

The multi-three-phase is the preferred configuration in many current applications because it resembles well-known three-phase machine topologies [18]. On the other hand, some critical points to consider that influence the winding arrangement of a nine-phase IM from a three-phase one are listed below:

- The power of the IM used is vital since, at higher power (from 5 HP), the dimension of slots is larger. They can therefore accommodate a more significant number of coils;

- The number of stator slots must be a multiple of the number of phases of the machine to be designed;
- The number of poles can be different from the initial one. Adding more poles will depend exclusively on the capacity of the machine and the chosen winding;
- The winding design will consider the one that presents minor difficulties for its arrangement and distribution inside the slots. This is at the discretion of the designer.

In Table 1, it can be seen which are the ideal values for the number of phases and poles proposed as a function of the number of slots the motor has, considering a simple arrangement (consequent poles to a layer) in the choice of the winding to be used.

**Table 1.** The number of poles $(2p)$ as a function of the number of slots $(K)$ and the number of phases $(q)$.

| Number of Slots (K) | Number of Phases (q) | | | |
|---|---|---|---|---|
| | 5 | 6 | 9 | 12 |
| 36 | x | 2 to 6 | 2 to 4 | 2 |
| 40 | 2 to 8 | x | x | x |
| 72 | x | 2 to 12 | 2 to 8 | 2 to 6 |
| 144 | x | 2 to 24 | 2 to 16 | 2 to 12 |

It can be seen in Table 1 that, for the same number of slots, by increasing the number of phases, the machine may have fewer poles, and this is due to the design limits that occur at the time of rewinding it. In the case of the design of a nine-phase IM, it is possible to make it with two poles and four poles in armatures whose number of slots is equal to 36. As for the design of one with two poles, there are more possibilities for more types of windings than the one with four poles, and the same would happen with one with six phases. As the number of poles increases, less space is available, and so is the design variety [19].

*2.2. Topology*

Multiphase motors can be arranged in two types: symmetrical and asymmetrical. Each one has its characteristics that are considered for the design of the winding of these machines. They can be AC or DC, synchronous or asynchronous, with a symmetrical or asymmetrical configuration between the phases. These machines are commonly designed in a symmetrical arrangement because it entails fewer difficulties for their control. It is necessary to mention that their control will be through frequency power converters [20].

2.2.1. Symmetrical

The sets can be arranged as symmetric if the spacial change between two consecutive phase windings is $\frac{2\pi}{q}$. Therefore, the set of phases for the case of a symmetrical nine-phase IM would be distributed by 40° between sets around the circumference of the stator as shown in Figure 1, maintaining the phase shift within each three-phase set at 120° electrical at all times [18].

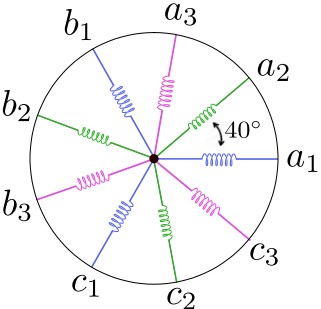

**Figure 1.** Symmetrical representation of a nine-phase motor with 40° electrical displacement between three-phase windings.

### 2.2.2. Asymmetrical

In the case of asymmetric nine-phase IMs, the three-phase windings are distributed in such a way that they are displaced by 20° between the phase windings and 120° between phases of the same assembly, as occurs in a motor with a symmetric configuration, as shown in Figure 2. Thus, the spacial change between the corresponding phases of the windings is $\frac{\pi}{q}$ [18].

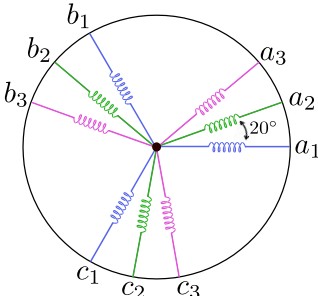

**Figure 2.** Asymmetrical representation of a nine-phase IM with 20° electrical displacement between the three-phase windings.

### 2.3. Calculations for AC Windings Applied to Nine-Phase IM

Two types of windings are usually used for the design and winding of electric motors: concentric and eccentric. At the same time, these are subdivided into concentric windings by consequent poles, concentric windings by poles, imbricated eccentric windings, and corrugated eccentric windings. Depending on the type, they can be made with single or two layers. They can be even, odd, or fractional integers [14].

A 380 V, 4 poles, 5 HP three-phase squirrel cage IM with 36 slots in the stator and 44 slots in the rotor was used in this paper. Some parameters were delimited for constructing the nine-phase motor winding:

- The power is equal to or close to that of a three-phase IM;
- The same number of poles is maintained;
- The windings are of asymmetric type.

Calculation Process

Known IM variables for calculations are:

- Number of slots: $K = 36$;
- Number of poles: $2p = 4$;
- Number of phases: $q = 9$.

1. First, the value of $K_{pq}$ is determined:

$$K_{pq} = \frac{K}{2pq} = \frac{36}{4 \times 9} = 1,\qquad(1)$$

being $K_{pq}$ an odd integer value. The design of windings by consequent poles is chosen. By taking into account the possibility of winding either a single layer or two layers, we proceed as follows:

2. Concentric windings:

- Single layer:

$$B = \frac{K}{2} = \frac{36}{2} = 18.\qquad(2)$$

- Two layers:

$$B = K = 36.\qquad(3)$$

where $B$ is the total number of coils. From here, calculations are developed in a single layer because it allows us to correctly take advantage of the available slots to distribute the coils.

3.  Total number of groups ($G$):

$$G = pq = 2 \times 9 = 18. \tag{4}$$

4.  The number of coils that make up each group ($U_g$) is calculated:

$$U_g = \frac{B}{G} = \frac{18}{18} = 1. \tag{5}$$

5.  Group width ($m$):

$$m = (q-1)K_{pq} = (9-1) \times 1 = 8. \tag{6}$$

6.  Finally, the distance between phases ($Y_q$) is determined:

$$Y_q = \frac{K\theta°}{360°p} = \frac{36 \times 20°}{360° \times 2} = 1. \tag{7}$$

As the calculation has now been carried out, the winding distribution for the new nine-phase IM with asymmetrical topology is presented in Figure 3:

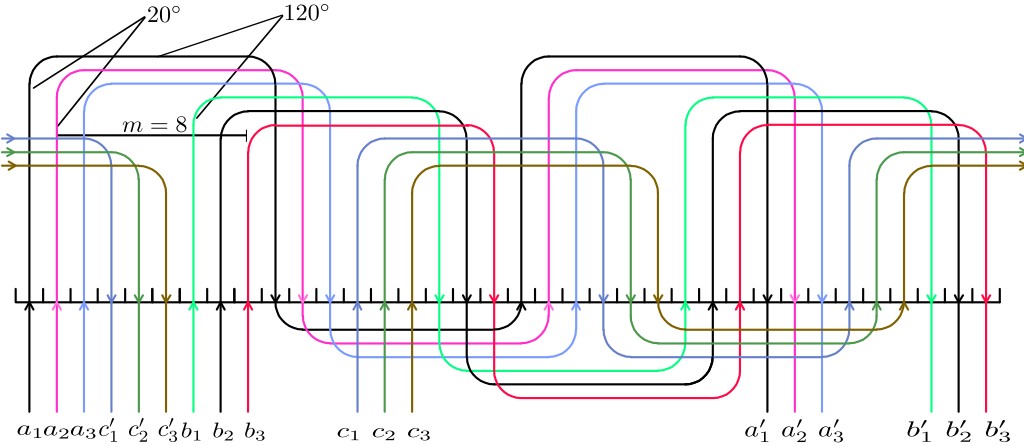

Nine-phase IM - Single layer consequent pole winding. $K = 36$, $2p = 4$, $q = 9$, $G_f = 2$, $U_g = 1$

**Figure 3.** Winding diagram by poles consequent to a single layer for a nine-phase IM with asymmetrical configuration.

## 3. Nine-Phase IM Analysis and Design

Based on the previous calculations for the windings design of the nine-phase IM, stator slots distribution was used. On the other hand, some parameters were obtained through physical measurements and the IM nameplate nominal values. Table 2 shows data on the IM nameplate, which are its corresponding nominal values.

**Table 2.** Data provided by the manufacturer attached to the IM casing.

| Parameter | Value |
| --- | --- |
| Voltage | 220/380 V |
| Nominal current | 17/9.8 A |
| Frequency | 50 Hz |
| Protection degree | IP54 |
| Speed | 1450 rpm |

Tables 3 and 4 present measured data from the mechanical dimensions of the stator and rotor, respectively. These values will be used in ANSYS simulations to obtain a precise IM model and to verify the nine-phase design later.

**Table 3.** Stator dimensions of the IM.

| Parameter | Value |
|---|---|
| Number of poles | 4 |
| Outer diameter | 180 mm |
| Inner diameter | 120 mm |
| Length | 120 mm |
| Number of stacked sheets | 171 |
| Sheet thickness | 0.95 mm |
| Slot number | 36 |
| Slot height $H_{s0}$ | 1 mm |
| Slot height $H_{s1}$ | 1 mm |
| Slot height $H_{s2}$ | 15 mm |
| Slot width $B_{s0}$ | 3 mm |
| Slot width $B_{s1}$ | 6 mm |
| Slot width $B_{s2}$ | 9 mm |

**Table 4.** Rotor dimensions of the IM.

| Parameter | Value |
|---|---|
| Outer diameter | 115 mm |
| Inner diameter | 42 mm |
| Length | 120 mm |
| Sheet thickness | 0.95 mm |
| Slot number | 44 |
| Slot height $H_{s0}$ | 2 mm |
| Slot height $H_{s1}$ | 1 mm |
| Slot height $H_{s2}$ | 12 mm |
| Slot width $B_{s0}$ | 2 mm |
| Slot width $B_{s1}$ | 3 mm |
| Slot width $B_{s2}$ | 6 mm |

*Preliminary Design*

Figure 4 illustrates the preliminary design of the IM's stator and rotor mechanical dimensions via ANSYS Maxwell software. Data from Tables 2–4 were used, as well as the winding layout presented in Figure 3, which include the stator and rotor slot shapes, to perform a preliminary design of the asymmetrical nine-phase IM. They were used in the virtual design environment ANSYS Maxwell 17.2 to obtain an approximate model of the studied IM.

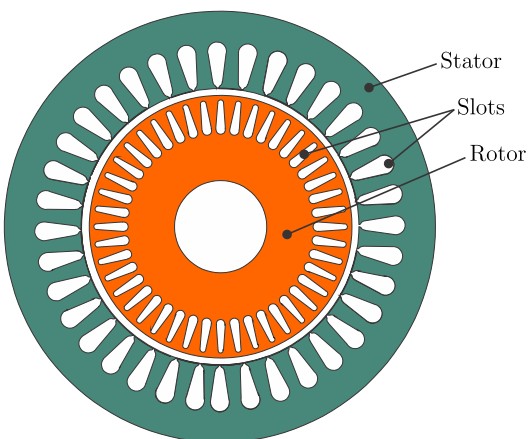

**Figure 4.** Preliminary design of the stator and rotor of an IM via ANSYS Maxwell 17.2.

The nine-phase design was considered a modified distribution to obtain an asymmetrical magnetic flux, as shown in Figures 5 and 6, where the phase shift in a steady state condition between adjacent three-phase windings is 20°, verifying the asymmetrical design of the nine-phase IM. The results for the three-phase IM are also shown.

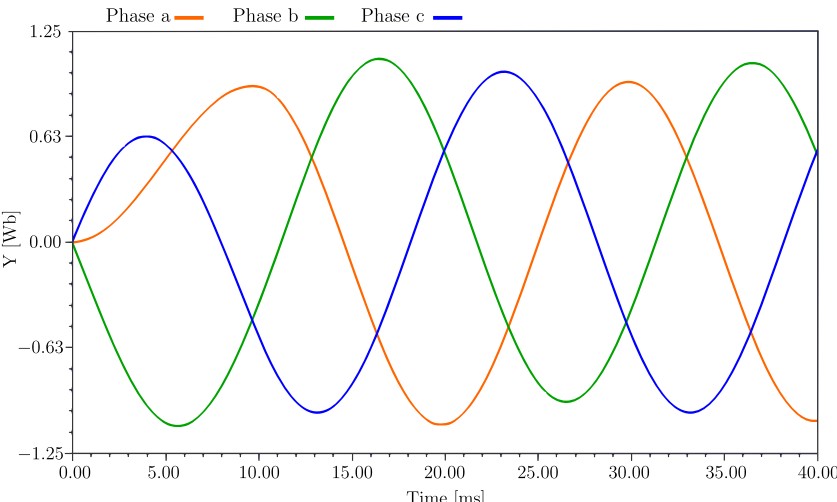

**Figure 5.** Three-phase winding plot fluxes.

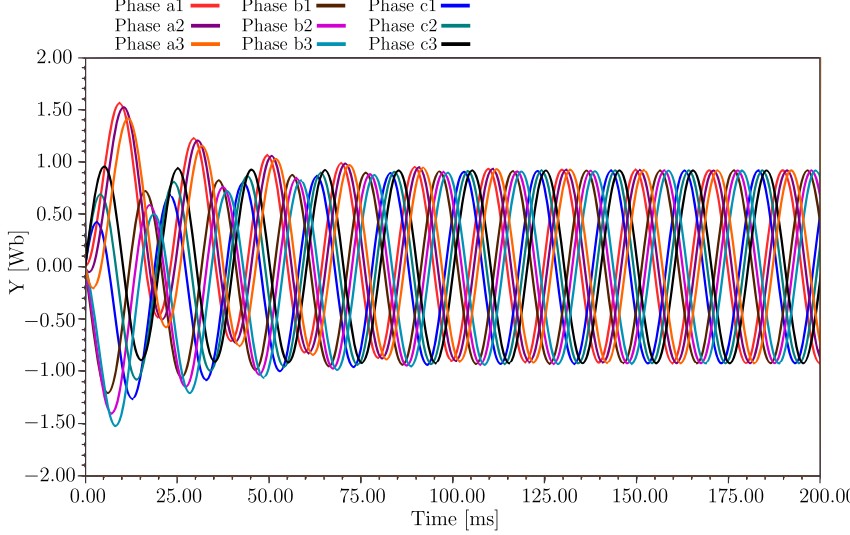

**Figure 6.** Nine-phase winding plot fluxes.

At the same time, a representation of the three-phase IM and asymmetrical nine-phase IM flux lines are presented in Figures 7 and 8, highlighting the number of poles generated through the flux lines.

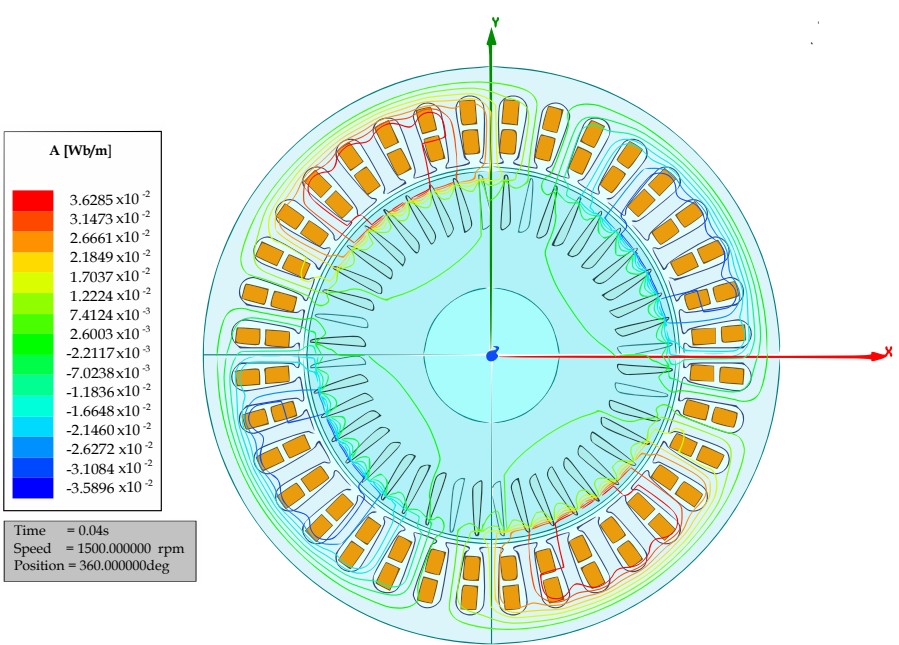

**Figure 7.** Magnetostatic modeling of a three-phase IM.

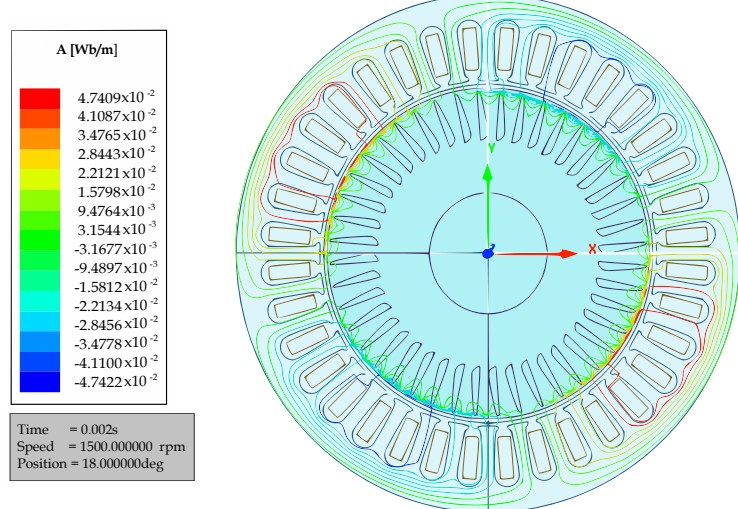

**Figure 8.** Magnetostatic modeling of a nine-phase IM.

## 4. Experimental Results

This section presents the results from tests carried out on the three-phase IM before being modified and on the nine-phase IM after being rewound to determine its performance. With these results, a comparison was made to point out their differences and similarities. Figure 9 depicts the experimental platform including the three-phase power source based on a varivolt, the oscilloscope, and the machine. Lastly, Figures 10 and 11 present pictures of the experimental setup to perform the parameters tests, including current and voltage probes.

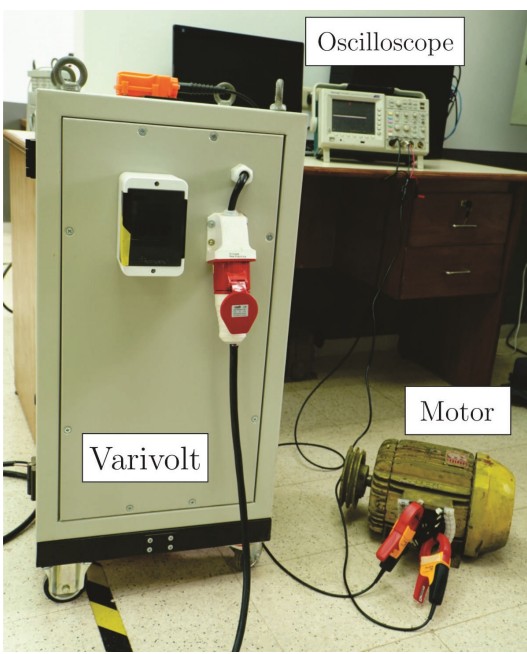

**Figure 9.** Experimental platform of the modified nine-phase IM.

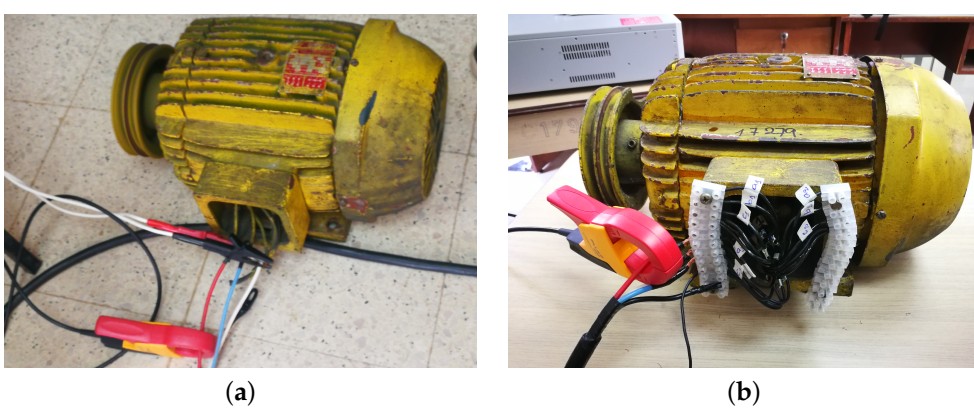

| (**a**) | (**b**) |

**Figure 10.** Experimental setup for non-load test. (**a**) Three-phase IM. (**b**) Nine-phase IM.

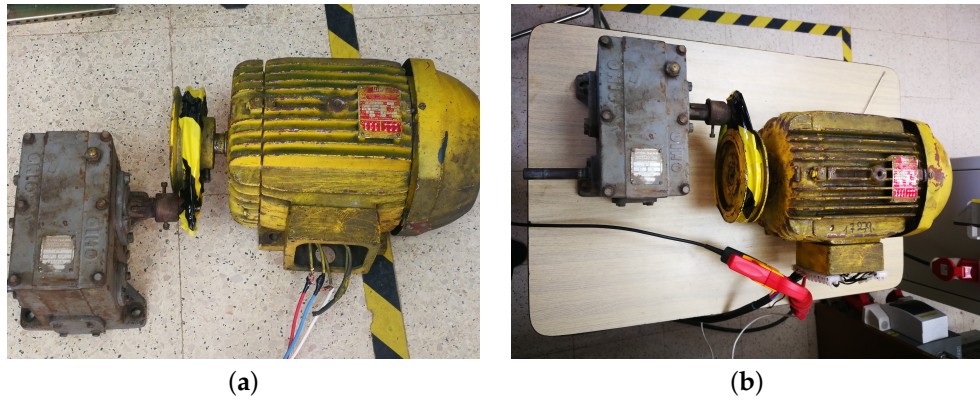

| (**a**) | (**b**) |

**Figure 11.** Experimental setup for locked rotor test. (**a**) Three-phase IM. (**b**) Nine-phase IM.

*4.1. Original Three-Phase IM Results*

Several tests were performed to obtain the three-phase IM's electrical parameters. These tests are named non-load test, locked-rotor test, and DC-test [21]. Table 5 presents the obtained values for these tests applied to the three-phase IM. Then, Table 6 shows

the electrical parameters for the three-phase IM, which were calculated through the obtained values.

**Table 5.** Obtained values through tests on the three-phase IM.

| Non-Load Test | Locked-Rotor Test | DC-Test |
|---|---|---|
| Line voltage 400 V | Applied voltage 63.3 V | Applied voltage 1.47 V |
| Nominal current 8 A | Current 10.2 A | Current 10.2 A |
| Angle 50.4° | Angle 18° | − |
| Speed 1480 rpm | − | − |

**Table 6.** Electrical parameters of the three-phase IM.

| Equivalent Circuit | |
|---|---|
| **Parameter** | **Value** |
| Stator resistance $R_s$ | 0.331 Ω |
| Rotor resistance $R_{r'}$ | 3.0767 Ω |
| Stator reactance $X_s$ | 3.0767 Ω |
| Rotor reactance $X_{r'}$ | 0.6642 Ω |
| Magnetising reactance $X_m$ | 36.795 Ω |

*4.2. Rewound Nine-Phase IM Results*

In this section, the results through the tests carried out on the new asymmetrical nine-phase IM are presented in Tables 7–9. The tests consist of the same performed for the three-phase IM. Measurements of the nine-phase IM were carried out independently for the three-phase sets. Electrical parameters of the nine-phase IM are described in Table 10.

**Table 7.** The obtained values of the non-load test carried out by independent sets at the asymmetrical nine-phase IM.

| Winding 1 | Winding 2 | Winding 3 |
|---|---|---|
| **Parameter and Value** | **Parameter and Value** | **Parameter and Value** |
| Phase voltage ($V_{a1-n1}$) 231 V | Phase voltage ($V_{a2-n2}$) 230 V | Phase voltage ($V_{a3-n3}$) 232 V |
| Current ($I_{a1}$) 1.26 A | Current ($I_{a2}$) 0.8 A | Current ($I_{a3}$) 1.27 A |
| Angle ($\phi_{a1}$) 70° | Angle ($\phi_{a2}$) 52° | Angle ($\phi_{a3}$) 66° |
| Power ($P_{a1}$) 99.5 W | Power ($P_{a2}$) 113.3 W | Power ($P_{a3}$) 119.8 W |

**Table 8.** Obtained values of the locked rotor test carried out by independent sets at the asymmetrical nine-phase IM.

| Winding 1 | Winding 2 | Winding 3 |
|---|---|---|
| **Parameter and Value** | **Parameter and Value** | **Parameter and Value** |
| Phase voltage ($V_{a1-n1}$) 51.7 V | Phase voltage ($V_{a2-n2}$) 51.8 V | Phase voltage ($V_{a3-n3}$) 51 V |
| Current ($I_{a1}$) 2.45 A | Current ($I_{a2}$) 2.53 A | Current ($I_{a3}$) 2.6 A |
| Angle ($\phi_{a1}$) 38.3° | Angle ($\phi_{a2}$) 34.7° | Angle ($\phi_{a3}$) 26.5° |
| Power ($P_{a1}$) 99.4 W | Power ($P_{a2}$) 107.7 W | Power ($P_{a3}$) 118.7 W |

**Table 9.** Obtained values of the DC-test carried out by independent sets at the asymmetrical nine-phase IM.

| Winding 1 | Winding 2 | Winding 3 |
|---|---|---|
| **Parameter and Value** | **Parameter and Value** | **Parameter and Value** |
| DC voltage ($V_{dc1}$) 24.1 V | DC voltage ($V_{dc2}$) 24.3 V | DC voltage ($V_{dc3}$) 26.5 V |
| DC current ($I_{dc1}$) 1 A | DC current ($I_{dc2}$) 1 A | DC current ($I_{dc3}$) 1 A |

**Table 10.** Electrical parameters of the asymmetrical nine-phase IM.

| Equivalent Circuit | |
| --- | --- |
| **Parameter** | **Value** |
| Stator resistance $R_s$ | 13.125 Ω |
| Rotor resistance $R_{r'}$ | 3.903 Ω |
| Stator reactance $X_s$ | 18.6 Ω |
| Rotor reactance $X_{r'}$ | 18.6 Ω |
| Magnetising reactance $X_m$ | 206.34 Ω |

At last, Figures 12 and 13 show the measured voltage and current in the oscilloscope for the non-load and locked rotor test of the nine-phase IM.

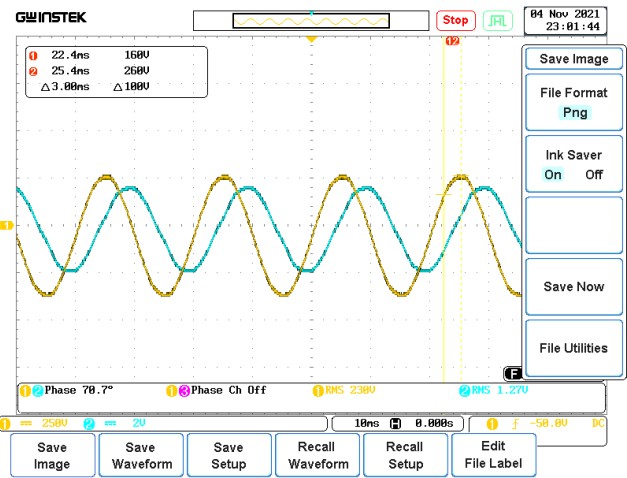

**Figure 12.** Experimental non-load test for winding 1 of the nine-phase IM including voltage (yellow line) and current (blue line).

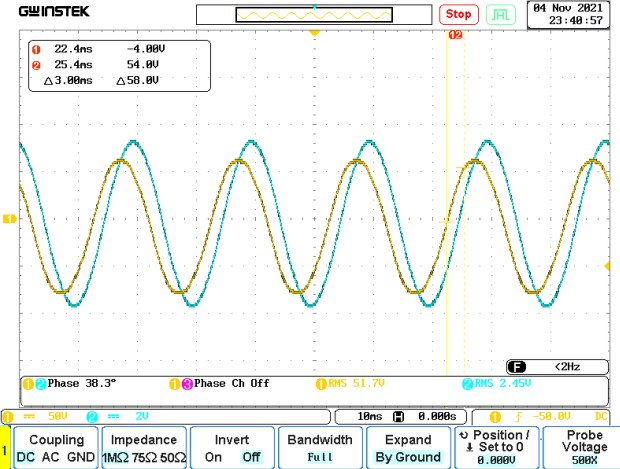

**Figure 13.** Experimental locked rotor test for winding 1 of the nine-phase IM including voltage (yellow line) and current (blue line).

### 4.3. Comparative Analysis-Discussion

This section discusses a comparative analysis between both designs, the three-phase IM and the rewound asymmetrical nine-phase IM. First, Figure 14 shows the equivalent circuit per phase for both IMs.

Figures 15 and 16 present a comparative view between both IMs in terms of power, phase current, and phase voltage.

Considering nominal currents for both three-phase and nine-phase IM, which are 9.8 A and 3.2 A, respectively, and the electrical parameters from the equivalent circuit from Figure 17, a similar condition in terms of efficiency can be estimated.

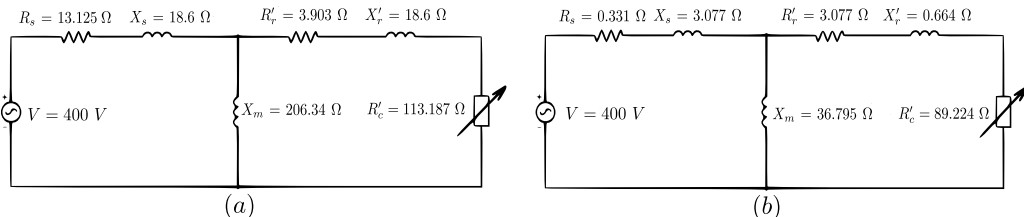

**Figure 14.** Equivalent circuit per phase. (**a**) Nine-phase IM. (**b**) Three-phase IM.

In nominal conditions, the three-phase IM consumes approximately 4500 VAR and has 1560 W power losses. On the other hand, the nine-phase IM consumes 3370 VAR and has 1280 W of power losses in total. This gives approximately 25% reactive power minimization and a reduction of 18% in terms of power losses.

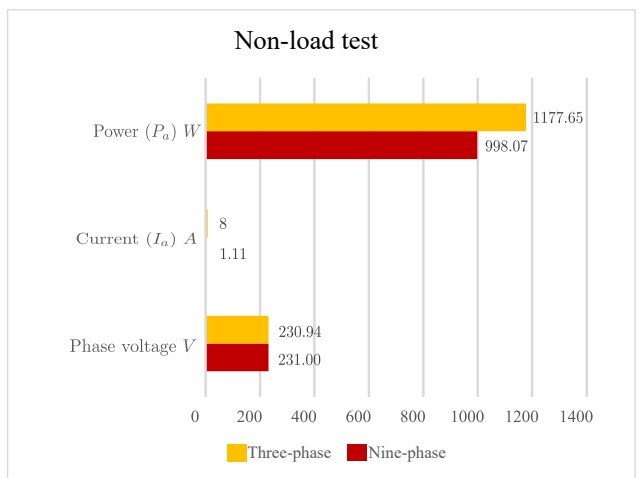

**Figure 15.** Comparison between the obtained values using the non-load test applied to the three-phase and nine-phase IM.

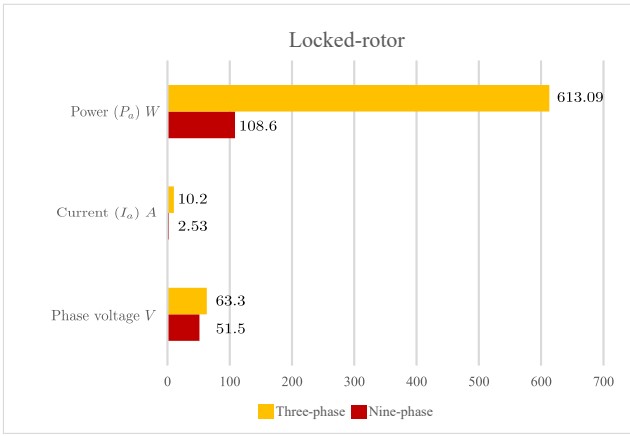

**Figure 16.** Comparison between obtained values using the locked-rotor test applied to the three-phase and nine-phase IM.

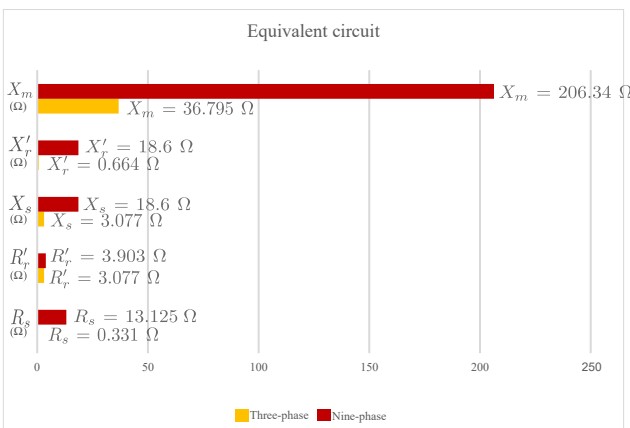

**Figure 17.** Comparison between the values of resistances and reactances of the equivalent circuit of both three-phase and nine-phase IM.

On the other hand, note that the nine-phase IM possesses higher leakage, higher magnetizing inductance, and higher resistance. Consequently, it acts as a low pass filter with a lower cutoff frequency, making it more versatile to power sources based on power electronics with high-frequency harmonics generating torque ripple-producing vibrations. It is worth noting that now that the IM has become a multiphase machine, it has gained the ability to post-fault operations, making it more suitable for critical applications.

## 5. Conclusions

This paper presents a winding design and efficiency analysis of a nine-phase IM. The versatility of the design is demonstrated by constructing the asymmetrical nine-phase IM from an original three-phase IM. The results obtained through tests applied to three-phase and nine-phase IM verified the following: currents through the windings of each phase are significantly reduced in the nine-phase IM, so the total reactive power consumption and power losses are reduced compared to the original three-phase IM, confirming a notorious increase in efficiency. Besides, leakage and magnetizing inductances are increased, obtaining a greater low pass filter for high-frequency harmonics currents produced by power electronics drives.

**Author Contributions:** Conceptualization, A.F. and M.A.; methodology, M.A. and O.G.; software, A.F. and M.A.; validation, A.F., M.A., O.G., L.D. and C.R.; formal analysis, A.F., M.A., O.G., L.D. and C.R.; investigation, A.F., M.A., O.G., L.D. and C.R.; resources, M.A. and J.R.; data curation, A.F. and O.G.; writing—original draft preparation, A.F., M.A., O.G., L.D., C.R. and J.R.; writing—review and editing, M.A., O.G., L.D., C.R. and J.R.; visualization, M.A. and J.R.; supervision, M.A.; project administration, M.A., J.R. and R.G.; funding acquisition, M.A., J.R. and R.G. All authors have read and agreed to the published version of the manuscript.

**Funding:** This research has been funded through the Facultad de Ingeniería, Universidad Nacional de Asunción.

**Institutional Review Board Statement:** Not applicable.

**Informed Consent Statement:** Not applicable.

**Data Availability Statement:** Not applicable.

**Acknowledgments:** Ayala, M., Gonzalez, O., Delorme, L., Romero, C., Rodas, J. and Gregor, R. acknowledge the support of CONACYT through its PRONII program.

**Conflicts of Interest:** The authors declare no conflict of interest.

**Abbreviations**

The following abbreviations are used in this manuscript:

| | |
|---|---|
| AC | Alternating Current |
| DC | Direct Current |
| DSD | Drive System Design |
| EDU | Electric Drive Unit |
| IM | Induction Motor |

**Nomenclature**

| | |
|---|---|
| $2p$ | Number of poles |
| $B$ | Total number of coils |
| $G$ | Total group of coils |
| $h$ | Number of isolated neutral points |
| $K$ | Number of slots |
| $K_{pq}$ | Number of slots per pole and phase |
| $m$ | Amplitude |
| $q$ | Number of phases |
| $U_g$ | Number of coil per group |
| $Y_q$ | Displacement between phase principles |

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
