# Peer review of "Winding Design and Efficiency Analysis of a Nine-Phase Induction Machine from a Three-Phase Induction Machine"

_machines, doi:10.3390/machines10121124_

Round 1

Reviewer 1 Report

This paper mainly studies the performance analysis of nine-phase induction motor. To introduce a straightforward procedure to rewind a nine-phase induction machine from a three-phase one. The study of the three-phase induction motor was performed, which included selecting a new winding design, calculating stator coils and simulation tests with ANSYS Maxwell software to validate the design. Experimental tests were also performed to obtain the electrical parameters of both windings and compare them in terms of power losses.

The research content of this paper is rich, with a certain degree of practical value and innovation. But the following changes are needed.

1The research content expressed by the thesis title is not clear.What kind of motor performance is studied in this paper?

2Chapter 2.1 and 2.2 analyzes Multi-three-phase winding. The "As the number of poles increases, less space is available, and so is the design variety " conclusion was reached. The "It is necessary to mention that their control will be through frequency converters" conclusion was reached. The "They can be made with single or two layers, depending on the type. They can be even, odd or fractional integers" conclusion was reached. But these conclusions have been put forward by other scholars, so where is the author's innovation?

3FIG. 5 and FIG. 6 are very fuzzy, and the data in them is not clear. The author is advised to redraw them. It is suggested to make a detailed comparative analysis of Figure 5 and Figure 5 to reflect the difference between the three-phase induction motor and the nine-phase induction motor.

4To ensure the authenticity of experimental data. Please add the pictures of the two motor prototypes and the pictures of the experimental platform in Chapter 4 of the paper.

5There are some spelling and grammar problems in the article. The author is advised to read it carefully.

Author Response

The authors would like to thank the reviewers and the editor for their favourable consideration of the work and their helpful comments and suggestions to improve it. In what follows, the reviewers provide a summary of changes on a point-by-point basis. Remarks of the reviewers are in italics and black font, while the authors’ reply is in standard red font in this document. The changes are identified in the new version of the manuscript using the red font.

Reviewer 2 Report

The article is very interesting and presents the problem of improving the performance of the induction motor most commonly used in the industry.

1) However, I have a doubt whether the use of a nine-phase system will not hinder the application of this solution due to the common use of three-phase inverters to power induction motors. Do you think that the complexity of the construction of the power supply system (inverter) does not eliminate the effects of the benefits achieved in the nine-phase motor?

2) Is the asymmetric rotor slot shape shown in Figures 4 and 6 purposeful ? If so, what is the purpose of introducing such a shape?

Author Response

(The authors gave the same response as above.)

Reviewer 3 Report

The topics raised by the authors are important and up-to-date, especially in the context of the parameters of polyphase motors. However, some things need to be fixed in the article.

1. Markings in figures 5 and 6 are written in too small font, which makes them unclear - I suggest using a larger font so that you do not have to enlarge the text when reading the article.

2. Chapter 4 lacks a more in-depth explanation of the results obtained - what these results mean for the machine's operation and whether it is good or whether the obtained parameters could be improved. The comparison concerns only reactive power and power losses, and there is no explanation of differences for other parameters.

3. Section 5 needs to be supplemented and corrected - here the authors should clearly indicate where the conclusions come from and on what basis they were formulated - e.g., the issue of a better filter for current harmonics.

Author Response

(The authors gave the same response as above.)

Round 2

Reviewer 1 Report

The author gives a detailed description of the suggestions for revision. In addition to the photos of the prototype, it is suggested to supplement the photos of the experimental platform. Please read through the text and check for spelling and grammar problems.

Author Response

The authors would like to thank the reviewers and the editor for their favourable consideration of the work and their helpful comments and suggestions to improve it. In what follows, the reviewers provide a summary of changes on a point-by-point basis. Remarks of the reviewers are in italics and black font, while the authors reply is in standard red font in this document. The changes are identified in the new version of the manuscript using the red font.
